# Effects of Non-Virion Gene Expression Level and Viral 
Genome Length on the Replication and Pathogenicity of Viral Hemorrhagic Septicemia Virus

**DOI:** 10.3390/v14091886

**Published:** 2022-08-26

**Authors:** Najib Abdellaoui, Seon Young Kim, Ki Hong Kim, Min Sun Kim

**Affiliations:** 1Department of Biological Sciences, Kongju National University, Gongju 32588, Korea; 2Department of Aquatic Life Medicine, Pukyong National University, Busan 48513, Korea

**Keywords:** recombinant VHSV, NV expression level, virus genome length, virus replication, type-I interferon, pathogenicity

## Abstract

Fish novirhabdoviruses, including viral hemorrhagic septicemia virus (VHSV), hirame rhabdovirus (HIRRV), and infectious hematopoietic necrosis virus (IHNV), harbor a unique non-virion (NV) gene that is crucial for efficient replication and pathogenicity. The effective levels and the function of the N-terminal region of the NV protein, however, remain poorly understood. In the present study, several recombinant VHSVs, which completely lack (rVHSV-ΔNV) or harbor an additional (rVHSV-dNV) NV gene, were generated using reverse genetics. To confirm the function of the N-terminal region of the NV protein, recombinant VHSVs with the NV gene that gradually mutated from the start codon (ATG) to the stop codon (TGA), expressed as N-terminally truncated NV proteins (rVHSV-NV1, -NV2, and -NV3), were generated. CPE progression and viral growth analyses showed that epithelioma papulosum cyprini (EPC) cells infected with rVHSV-ΔNV or rVHSV-NV3—which did not express NV protein—rarely showed CPE and viral replication as opposed to EPC cells infected with rVHSV-wild. Interestingly, regardless of the presence of two NV genes in the rVHSV-dNV genome, EPC cells infected with rVHSV-dNV or rVHSV-A-EGFP (control) failed to induce CPE and viral replication. In EPC cells infected with rVHSV-dNV or rVHSV-A-EGFP, which harbored a longer VHSV genome than the wild-type, Mx gene expression levels, which were detected by luciferase activity assay, were particularly high; Mx gene expression levels were higher in EPC cells infected with rVHSV-ΔNV, -NV2, or -NV3 than in those infected with rVHSV-wild or rVHSV-NV1. The total amount of NV transcript produced in EPC cells infected with rVHSV-wild was much higher than that in EPC cells infected with rVHSV-dNV. However, the expression levels of the NV gene per viral particle were significantly higher in EPC cells infected with rVHSV-dNV than in cells infected with rVHSV-wild. These results suggest that the NV protein is an essential component in the inhibition of host type-I interferon (IFN) and the induction of viral replication. Most importantly, viral genome length might affect viral replication efficiency to a greater extent than does NV gene expression. In in vivo pathogenicity experiments, the cumulative mortality rates of olive flounder fingerlings infected with rVHSV-dNV or rVHSV-wild were similar (60–70%), while those of fingerlings infected with rVHSV-A-EGFP were lower. Moreover, the virulence of rVHSV-ΔNV and rVHSV, both harboring a truncated NV gene (rVHSV-NV1, -NV2, and -NV3), was completely attenuated in the olive flounder. These results suggest that viral pathogenicity is affected by the viral replication rate and NV gene expression. In conclusion, the genome length and NV gene (particularly the N-terminal region) expression of VHSVs are closely associated with viral replication in host type-I IFN response and the viral pathogenicity.

## 1. Introduction

Viral hemorrhagic septicemia virus (VHSV)—a member of the genus, *Novirhabdovirus*, of the family, *Rhabdoviridae*—causes severe viral hemorrhagic septicemia in freshwater and seawater fishes worldwide. The VHSV genome comprises approximately 11 kb of negative-sense single-stranded RNA and encodes 5 structural proteins: nucleoprotein (N), polymerase-associated phosphoprotein (P), matrix protein (M), glycoprotein (G), and RNA-dependent RNA polymerase (L). Unlike that of mammalian rhabdoviruses, the RNA genome of fish rhabdoviruses harbors a unique, small, non-structural gene encoding a non-virion protein (NV). Many recent studies have attempted to uncover the roles of this NV gene in recombinant viruses generated via reverse genetics. In snakehead rhabdovirus (SHRV), the absence or presence of the NV gene did not affect its pathogenicity or replication [1,2]. However, in infectious hematopoietic necrosis virus (IHNV) and VHSV, the NV gene plays essential roles in virus growth and pathogenicity [3,4,5,6]. In our previous studies, we generated recombinant VHSV that lacked the NV gene and reported that the NV-knockout VHSV suppressed viral replication and virulence [7,8]. Furthermore, NV-knockout VHSV upregulated host type-I interferon (IFN) response—the first-line defense against viral infection in most vertebrates [9]. Biacchesi et al. (2017) demonstrated the inhibitory effects of fish Novirhabdovirus NV proteins and PPM1Bb protein phosphatase on RIG-I- and TBK-1-dependent IFN and IRF-stimulated gene induction in fish cells, resulting in the establishment of an innate antiviral state [10]. The N-terminal region between the first and second α-helices (the hinge region) of the NV gene is vital for IFN system modulation and viral pathogenesis [11]. In the present study, to investigate the associations of VHSV NV protein expression and genome length with viral growth, host type-I IFN response, and pathogenicity, we generated several recombinant VHSVs that either lacked or had an additional NV gene in their genomes. Furthermore, to confirm the role of the N-terminal region of the NV protein, recombinant VHSVs were generated from gradually mutating the NV gene by exchanging the start codon (ATG) with the stop codon (TGA), which expressed the N-terminally truncated NV proteins, and analyzed the effects of these truncations on viral growth, host type-I IFN response, and pathogenicity.

## 2. Materials and Methods

### 2.1. Cells and Viruses

Epithelioma papulosum cyprini (EPC) cells, which were high-passage and low-passage, were cultured in Leibovitz medium (L-15, Sigma) supplemented with penicillin (100 U/mL), streptomycin (100 mg/mL), and 10% fetal bovine serum (FBS, Sigma) [12]. EPC cells that contained pOFMx (olive flounder Mx1) reporter vector or those expressing T7 RNA polymerase were selected (in our previous experiment [8,9]) for G-418 (400 mg/mL, Sigma) supplementation. We propagated the previously recovered recombinant virus, rVHSV-wild [8], in a monolayer of high-passage EPC cells at 15 °C in the presence of 2% FBS and antibiotics (penicillin and streptomycin).

### 2.2. Vector Construction for the NV Gene-Mutated Recombinant VHSVs Generation

First, to generate recombinant VHSVs that overexpressed the NV gene and expressed the EGFP gene as a control, our previously constructed plasmid, pVHSV-A-RFP, was used [13]. To insert the NV or EGFP gene between the N and P genes of the VHSV genome, we amplified each gene containing the *Nde*I and *Sal*I sites using PCR with NV ORF-*Nde*I-F and NV ORF-*Sal*I-R or EGFP ORF-*Nde*I-F and EGFP ORF-*Sal*I-R (Table 1). The thermal cycling conditions were as follows: 1 cycle of 3 min at 95 °C (initial denaturation), followed by 30 cycles of 30 s at 95 °C, 30 s at 60 °C, and 30 s at 72 °C, with a final extension step of 7 min at 72 °C. Each product was cloned in pGEM T Easy Vector (Promega) and sequenced. The NV and EGFP ORFs were excised using *Nde*I and *Sal*I and inserted into the pVHSV-A-RFP vector in which the RFP ORF was digested using the same restriction enzymes, resulting in pVHSV-dNV and pVHSV-A-EGFP.

Second, to generate the recombinant VHSV that completely lacked the NV gene, the 5′ end of the NV ORF in the vector (pVHSV-wild) was altered to harbor a *Nar*I site using a site-directed mutagenesis kit (SDM, Stratagene) and named “pVHSV-wild SDM”. The primers used for SDM are in Table 1. The NV ORF in the rVHSV-wild SDM was removed by digestion with *Nar*I, and the digested fragment was self-ligated to produce pVHSV-ΔNV.

Third, to generate the recombinant VHSV-expressing truncated NV proteins, we changed the ATG codons in the NV gene to TGA codons using SDM (Stratagene). The primers used for SDM are listed in Table 1. Briefly, we constructed the plasmid, pVHSV-NV1, by changing the first ATG to TAG using pVHSV-wild as a template; pVHSV-NV2 by the mutagenesis of both the first and second ATGs to TGA with pVHSV-NV1 as a template; and pVHSV-NV3 by the mutagenesis of the first, second, and final ATGs to TGA using pVHSV-NV2 as a template. For SDM, we used the following thermal cycling conditions: 1 cycle of 3 min at 95°C, followed by 18 cycles of 30 s at 95 °C, 1 min at 55 °C, and 14 min at 68 °C. Finally, we applied a treatment with a *Dpn*I restriction enzyme to obtain the mutated plasmids.

In Figure 1, we give the constructed vectors in this study for the generation of NV gene-mutated recombinant VHSVs.

### 2.3. Rescue and Titration of Recombinant VHSVs

EPC cells that express T7 RNA polymerase were seeded in 6-well plates at 28 °C in L-15 medium supplemented with 20 mM HEPES and 10% FBS. Cells were cultured to ~80% confluence and co-transfected with 2 μg of each of the viral vectors and the supporting plasmids [pCMV-N (500 ng), -P (300ng), -L (200 ng)] using Fugene HD (Promega, Madison, WI, USA) according to the manufacturer’s instructions. Transfected cells were incubated for 12 h at 28 °C and the temperature was reduced to 15 °C. When total CPE was observed, the viral supernatant was harvested, filtered with a 0.45 μm syringe filter, and stored at −80 °C. We named the resulting viruses, “rVHSV-dNV, rVHSV-A-EGFP, rVHSV-ΔNV, rVHSV-NV1, rVHSV-NV2, and rVHSV-NV3”, respectively.

After three serial passages, we performed a plaque assay to quantify the infectious viral particles. EPC cell monolayers (1.5 × 10^6^ cells/35-mm dish) were inoculated with serially-diluted recombinant VHSVs and incubated at 15 °C for 1 h. The inoculum was removed and the cells were overlaid with plaquing medium (0.8% agarose in L-15 containing 2% FBS and antibiotics). After 7 days of incubation, the cells were fixed by 10% formalin and stained with 5% crystal violet for 30 min at room temperature. After rinsing the cells with distilled water, the plaque-forming units (PFU) were counted.

### 2.4. Effect of NV Gene on Viral Replication

EPC cell monolayers were seeded in 6-well plates (1.5 × 10^6^ cells/well) at 15 °C in L-15 medium containing 2% FBS and antibiotics and then infected with either recombinant VHSVs, rVHSV-wild, rVHSV-A-EGFP, rVHSV-dNV, rVHSV-ΔNV, rVHSV-NV1, rVHSV-NV2, or rVHSV-NV3 at MOI 0.1. At 1-, 3-, 5-, and 7-days post-infection, we observed CPE and analyzed each virus titer using the plaque assay

### 2.5. Effect of NV Gene on Type-I Interferon Response

EPC cells harboring olive flounder Mx1 (pOFMx) reporter vector were cultured in 6-well plates (1.5 × 10^6^ cells/well) at 15 °C and infected with the recombinant VHSVs, rVHSV-wild, rVHSV-A-EGFP, rVHSV-dNV, rVHSV-ΔNV, rVHSV-NV1, rVHSV-NV2, or rVHSV-NV3 at MOI 0.1. At 24-h post-infection, luciferase activity was measured using VICTOR3 (PerkinElmer, Waltham, MA, USA) according to the manufacturer’s instructions.

### 2.6. Quantification of the NV Gene Expression in EPC Cells Using Real-Time RT-PCR

EPC cells were cultured in 6-well plates (1.5 × 10^6^ cells/well) at 15 °C and were infected with either the recombinant VHSVs, rVHSV-wild, or rVHSV-dNV at MOI 0.1. At 12-, 24-, and 48-h post-infection, the total RNA was extracted from the infected EPC cells (rVHSV-wild or rVHSV-dNV) using a Hybrid-R Kit (Gene All, Korea), and 1 μg of total RNA was used to synthesize cDNA using a HyperScript First Strand Synthesis Kit with oligo (dT) primer or random primer (Gene All) according to the manufacturer’s instructions. We performed real-time RT-PCR using a Light Cycler 480 (Roche) and constructed each vector to contain a G-NV region or an NV gene for absolute quantification. cDNA synthesized with a random primer was used to quantify each virus genome, and cDNA synthesized with oligo(dT) was used as a template to amplify the NV gene transcript alone, whose primers are in Table 1. We ran the PCR reactions in a 20 μL volume using 2× SYBR Green Premix (Enzynomics) with 5 μL of cDNA and 5 pM of each primer with the thermal cycling conditions: 1 cycle of 15 min at 95 °C (pre-incubation), followed by 40 cycles of 10 s at 95 °C, 10 s at 60 °C, and 20 s at 72 °C.

### 2.7. In Vivo Pathogenicity of Recombinant VHSVs

The olive flounder fingerlings (4–5 g) that were pre-confirmed as free from VHSV and showed no signs of illness were infected by intramuscular (IM) injection of either the wild-type VHSV KJ2008, rVHSV-wild, rVHSV-A-EGFP, rVHSV-dNV, rVHSV-ΔNV, rVHSV-NV1, rVHSV-NV2, or rVHSV-NV3 at 10^3^ PFU per fish; control fish were injected with an equal volume of L-15 alone. We kept 10 fish in each group in nine 30 L tanks of two replicates at 14 °C and recorded the cumulative mortalities daily for 25-days post-injection.

### 2.8. Statistical Analysis

The data were analyzed for statistical significance with Graph Prism 5.0 software (GraphPad Prism, San Diego, CA, USA) using one-way ANOVA, with a *p* < 0.05 being considered statistically significant. The whole study was based on three independent experimental replicates.

## 3. Results

### 3.1. Propagation and Titration of Recombinant VHSVs

The recombinant VHSVs that were rescued in this study were also efficiently propagated by over three serial passages in high-passage EPC cells [12]. The titer of the rVHSV-wild was 1.9 × 10^7^ PFU/mL; (a) rVHSV-ΔNV was 3.3 × 10^6^ PFU/mL; (b) rVHSV-NV1 was 1.8 × 10^6^ PFU/mL; (c) rVHSV-NV2 was 8.0 × 10^5^ PFU/mL; (d) rVHSV-NV3 was 6.0× 10^5^ PFU/mL; (e) rVHSV-dNV was 5.5 × 10^5^ PFU/mL; and rVHSV-A-EGFP was 9.8 × 10^5^ PFU/mL.

### 3.2. Effects of the NV Gene on Viral Replication

The CPE progression in EPC cells infected with rVHSV-wild was clearly more extensive and faster than that in cells infected with rVHSV-ΔNV. Among rVHSV-NV1, -NV2, and -NV3, we didn’t observe any CPE in rVHSV-NV3, and -NV2 showed a less extensive CPE than -NV1. Moreover, the recombinant viruses, rVHSV-A-EGFP and rVHSV-dNV, didn’t cause any CPE in EPC cells (Figure 2a).

At all analyzed times, rVHSV-ΔNV growth was significantly lower than rVHSV-wild growth. Among rVHSV-NV1, -NV2, and -NV3, the titer of rVHSV-NV1 was considerably higher than those of rVHSV-NV2 and -NV3. Additionally, at different points in time, both rVHSV-A-EGFP and rVHSV-dNV showed no growth (Figure 2b).

### 3.3. Effects of the NV Gene on Type-I INF Response

The type-I IFN response in EPC cells infected with rVHSV-ΔNV, -NV2, or -NV3 was significantly higher than that in those infected with rVHSV-wild. Additionally, EPC cells infected with rVHSV-A-EGFP or rVHSV-dNV showed a considerably increased type-I IFN response than those infected with rVHSV-wild (Figure 3).

### 3.4. Quantification of NV Gene Expression in EPC Cells Using Real-Time RT-PCR

The total amount of NV transcripts produced in EPC cells infected with rVHSV-wild was much higher than in those infected with rVHSV-dNV (Figure 4a). However, the NV gene expression levels per genome copy were higher in EPC cells infected with rVHSV-dNV than in those infected with rVHSV-wild at 12- and 24-h post-infection (Figure 4b).

### 3.5. In Vivo Pathogenicity of Recombinant VHSVs

The cumulative mortality rate of olive flounder fingerlings i.m. injected with the wild-type VHSV KJ2008 was highest at 80–100%. Fish challenged with the rVHSV-dNV or rVHSV-wild showed similar mortality rates of 60–70%, while fish challenged with rVHSV-A-EGFP showed significantly decreased cumulative mortality rates. Moreover, fish challenged with the rVHSV-ΔNV, rVHSV-NV1, rVHSV-NV2, or rVHSV-NV3 showed no mortality (Figure 5).

## 4. Discussion

Several recent studies regarding the roles of the NV gene in *Novirhabdovirus* replication and host IFN response have used recombinant viruses generated via reverse genetics [1,2,3,4,5,6,7,8,9,14,15]. However, knowledge of how the NV protein functions in novirhabdoviruses remains limited. In the present study, we generated several NV-mutated recombinant VHSVs. First, we generated rVHSV-dNV harboring an additional NV gene between the N and P genes of the VHSV genome. We also generated rVHSV-ΔNV, which completely lacked the NV gene. Finally, we generated rVHSV-NV1, -NV2, and -NV3, which contained a gradually-mutated NV gene with the start codon (ATG) replaced by the stop codon (TGA). Compared to EPC infected with rVHSV-wild, CPE progression was rarely observed in EPC cells that were infected with both types that did not express the NV protein. This observation may be explained by the function of NV during VHSV infection. The NV gene induces downregulation of immune-related signaling pathways (RIG-I-like, and NOD-like signaling, and NF-kappa B signaling pathways), which allow viral replication and therefore CPE progression in cells infected with VHSV [16]. Therefore, the knockout of the NV gene led to a CPE that rarely progresses. Chinchilla et al. (2015) demonstrated that the injection of rainbow trout with recombinant NV induced the downregulation of several genes related to the innate and adaptive immune systems [17]. Among the downregulated genes, there were tumor necrosis factors, toll-like receptors, antigen processing and presentation, and interleukins. In addition, the deletion of NV gene in rIHNV severely inhibited the growth of the virus, which was restored using fish cell lines that expressed the NV protein [3]. Furthermore, the deletion of NV from the viral genome (rVHSV-ΔNV-EGFP) induced a higher NF-κB activity in comparison to the wild type VHSV, suggesting that VHSV evolved a tactic to suppress the early activation of NF-κB [18].

On the other hand, EPC cells infected with rVHSV-dNV or with rVHSV-A-EGFP did not show any CPE in comparison to control EPC cells infected with rVHSV-wild. Previously, we demonstrated that insertion of a foreign gene into the anterior part of VHSV genome led to a decrease in the virus’s replication, which suggests that the length of the genome may play a critical role in viral replication efficiency [19]. Based on our results, the lack of CPE in cells infected with rVHSV-dNV or rVHSV-A-EGFP could be caused by the lengthening of the viral genome, which causes a decrease in viral replication.

Concerning mutations in the NV genes, rVHSV-NV1 and rVHSV-NV2 induced higher CPE in comparison to rVHSV-NV3 (Figure 2a). Chinchilla et al. (2017) demonstrated the importance of the N-terminal of NV in viral replication through the inhibition of the MX and IL8 genes [11]. Furthermore, the C-terminal of NV (fragment2) upregulated the MX and IL8 genes compared to the other fragments [11]. In our study, the mutation of methionine at position 1 (NV1) had no effect on the function of the N-terminus for the suppression of the host type-I IFN response (Figure 3). However, the mutation of methionine at two positions (1 and 20) abolished the inhibitory role of NV on the type-I IFN response (NV2). These results suggest that the protein’s structure is important in the function of NV since the first mutation (NV1) did not abolish the inhibition of the NV gene for the type-I IFN response.

To quantify NV expression levels in EPC cells infected with rVHSV-dNV and rVHSV-wild, the total amount of NV transcripts and NV gene expression levels per virus particle were compared. The total amount of NV transcripts produced in EPC cells infected with rVHSV-wild was much higher than those in those infected with rVHSV-dNV (Figure 4a). However, the NV gene expression levels per genome copy were higher in the EPC cells infected with rVHSV-dNV than in those infected with rVHSV-wild (Figure 4b). Therefore, EPC cells infected with rVHSV-dNV showed lower replication efficiency, lower NV gene expression levels, and a higher host type-I IFN response. These results suggest that an extra NV gene in the viral genome didn’t significantly increase the NV transcript levels and showed a low virus titer in comparison to the rVHSV-wild. Therefore, the insertion of a new gene (genome size increase) will cause a reduction in the level of downstream genes and induce a reduction in virus replication and virus titer [20].

Previous studies demonstrated that lengthening the genome size reduced viral replication. Shin et al. (2000) showed that lengthening the genome of Moloney Murine Leukemia Virus resulted in a 5–10-fold reduction in viral replication compared to native genomes [21]. In addition, insertion of the EGFP gene into the VHSV genome reduced the replication efficiency in comparison to the wild-type virus [19].

In in vivo studies of the pathogenicity of recombinant viruses in olive flounder fingerlings, fish infected with rVHSV-dNV or rVHSV-wild showed similar morality rates of 60–70%; however, fish infected with rVHSV-A-EGFP showed significantly lower mortality rates (Figure 5). In our previous study, the insertion of a foreign gene (RFP or IFNγ) between the N and P genes resulted in a lower mortality rate than the wild-type VHSV [22]. Furthermore, the viral titer was lower for rVHSV-Arfp and rVHSV-A-IFNγ in comparison to the wild-type. These results suggest that viral genome length affects viral replication and pathogenicity, and reduces the pathogenicity and lowers viral replication (rVHSV-dNV and rVHSV-EGFP).

On the other hand, rVHSV-ΔNV and rVHSV harboring the truncated NV gene (rVHSV-NV1, -NV2, -NV3) displayed completely attenuated viral virulence in olive flounder. Ammayappan et al. (2011) demonstrated that the conversion of the start codon and internal ATG of NV ORF into a stop codon reduced the anti-apoptotic activity of the NV gene, resulting in a lower viral titer in comparison to wild-type rVHSV [4]. Moreover, knockout of NV and the introduction of a stop codon along the NV ORF produced a non-functional NV protein that led to an increase of type-I IFN response in comparison with the wild-type [16]. In addition, knockout of NV induced more trout immune genes than the wt-VHSV and upregulated several signaling pathways (RIG-I (retinoic acid-inducible gene-I) like receptor signaling, Toll-like receptor signaling, type-II interferon signaling, and nuclear factor kappa B (NF-kappa B) signaling) which indicates the wild-type for the delayed activation of antiviral, pro-inflammatory, and apoptosis pathways [16]. Furthermore, the mutation of a single amino acid in the NV gene caused an attenuation of virus virulence and showed higher mortality rates (80%) in trout [15]. Thus, rVHSV-ΔNV, rVHSV-NV1, -NV2, and -NV3 didn’t show high mortality in accordance with prior studies [15,23].

These results suggest that the NV gene of VHSV is a critical factor for viral pathogenicity and plays an important role during the first stage of replication via downregulation of several immune-related genes.

In conclusion, we generated several recombinant VHSVs associated with NV gene expression and analyzed the association of NV gene expression and viral genome length with viral replication and pathogenicity. We showed that the N-terminal region of the NV gene has a pivotal role in viral replication, host type-I IFN response, and viral pathogenicity. Moreover, the insertion of a new gene into the genome can lead to a low viral titer.

## Figures and Tables

**Figure 1 viruses-14-01886-f001:**
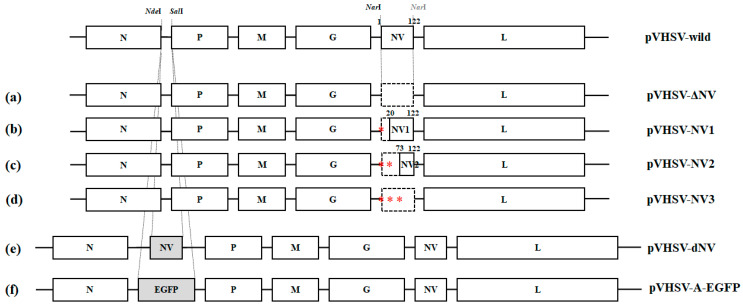
Construction of vectors for NV gene-mutated recombinant VHSVs. (**a**) pVHSV-ΔNV completely lacking the NV gene from the viral genome and (**b**) pVHSV-NV1, (**c**) -NV2, and (**d**) -NV3 harboring the truncated NV proteins by mutagenesis of the first; first and second; or first, second, and third ATG (M, methionine) codons to TAG codons, respectively. (**e**) pVHSV-dNV harboring an additional NV gene between the N and P genes of the viral genome and (**f**) pVHSV-A-EGFP as a control, harboring the EGFP reporter gene in the same region. * *p*  <  0.05, ** *p*  <  0.01, *** *p*  <  0.001.

**Figure 2 viruses-14-01886-f002:**
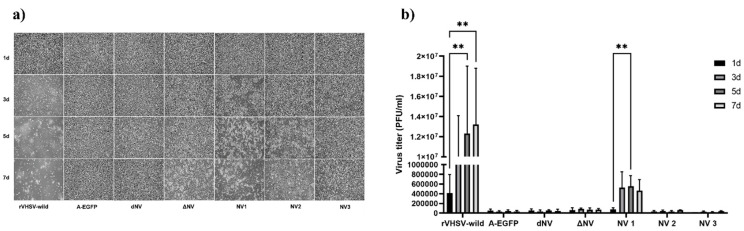
(**a**) CPE progression in EPC cells infected with rVHSV-wild, rVHSV-A-EGFP (A-EGFP), rVHSV-dNV (dNV), rVHSV-ΔNV (ΔNV), rVHSV-NV1 (NV1), rVHSV-NV2 (NV2), or rVHSV-NV3 (NV3) at MOI 0.1. (**b**) At 1-, 3-, 5-, and 7-days post-infection, we analyzed the growth of each virus using plaque assay (*n* = 3, means ± SD, ** *p*  <  0.01).

**Figure 3 viruses-14-01886-f003:**
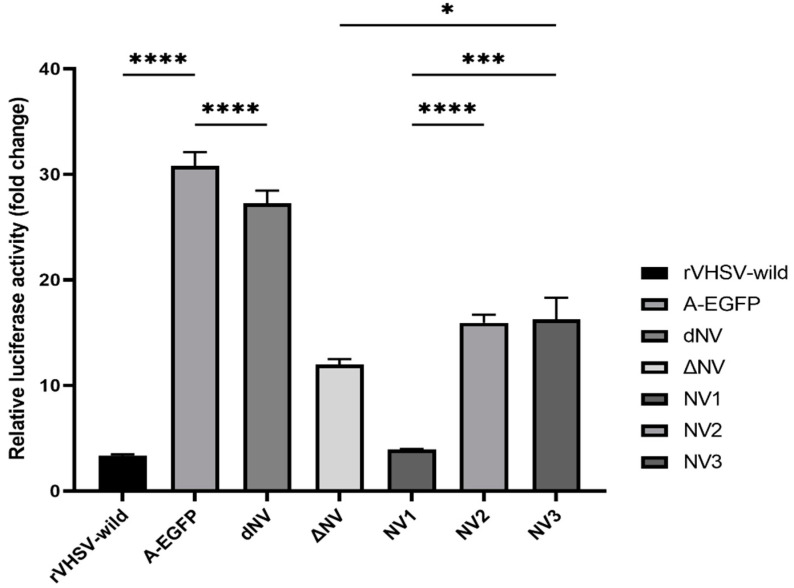
Analysis of Mx1 gene expression in EPC cells measured using luciferase activity. We infected EPC cells with rVHSV-wild, rVHSV-A-EGFP (A-EGFP), rVHSV-dNV (dNV), rVHSV-ΔNV (ΔNV), rVHSV-NV1 (NV1), rVHSV-NV2 (NV2), or rVHSV-NV3 (NV3) at MOI 0.1 and assessed the luciferase activity at 24-h post-infection (*n* = 6, means ± SD, * *p*  <  0.05, *** *p*  <  0.001, **** *p* < 0.0001).

**Figure 4 viruses-14-01886-f004:**
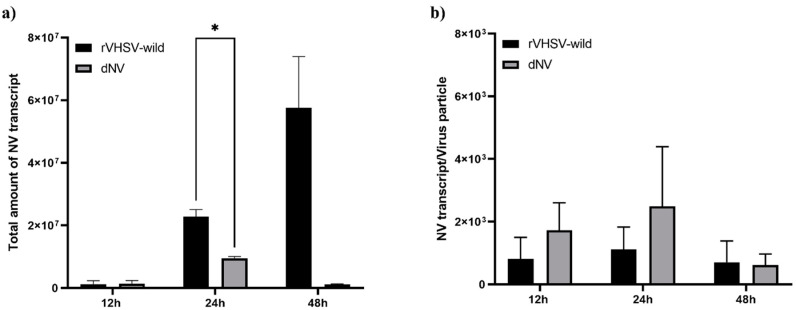
Quantification of the total number of NV transcripts (**a**) and NV gene expression levels per virus particle (**b**) in EPC cells using real-time RT–PCR. We inoculated EPC cells with rVHSV-wild or rVHSV-dNV (dNV) at MOI 0.1 and quantified the number of NV transcripts at 12-, 24-, and 48-h post-infection (*n* = 3, means ± SD, * *p*  <  0.05).

**Figure 5 viruses-14-01886-f005:**
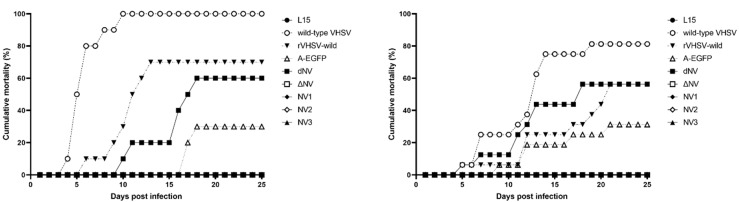
Cumulative mortality rates of olive flounder fingerlings challenged with wild-type VHSV, rVHSV-wild, rVHSV-A-EGFP (A-EGFP), rVHSV-dNV (dNV), rVHSV-ΔNV (ΔNV), rVHSV-NV1 (NV1), rVHSV-NV2 (NV2), or rVHSV-NV3 (NV3) at 10^3^ PFU per fish (2 replicates). We recorded the cumulative mortality rates daily for 25 days post-injection.

**Table 1 viruses-14-01886-t001:** Summary of primers used in this study.

Name of Primer	Sequence (5′ to 3′)	
NV ORF-*Nde*I	F	**CATATG**ATGACGACCCAGTCGGCAC
NV ORF-*Sal*I	R	**GTCGAC**TCATGGGGGAGATTCGGAG
EGFP ORF-*Nde*I	F	**CATATG**GTGAGCAAGGGCGAGGAGC
EGFP ORF-*Sal*I	R	**GTCGAC**TTACTTGTACAGCTCGTCCA
NV 5′ *Nar*I SDM	F	GCACCCCTGTGAGAC**GGCGCC**ATGACGACCCAGTCGGCACAC
NV 5′ *Nar*I SDM	R	GTGTGCCGACTGGGTCGTCAT**GGCGCC**GTCTCACAGGGGTGC
NV1-TGA SDM	F	GTGAGACATCGATTGAACGACCCAGTCGGCACACAG
NV1-TGA SDM	R	CTGTGTGCCGACTGGGTCGTTCAATCGATGTCTCAC
NV2-TGA SDM	F	TTCTCTCCACTTGTCCTTCGCGAGTGAATCGAGTACAGACTAACATTTG
NV2-TGA SDM	R	CAAATGTTAGTCTGTACTCGATTCACTCGCGAAGGACAAGTGGAGAGAA
NV3-TGA SDM	F	CCCTACCTCCATGTGCTTGACTGAAGGATAAGTCTCCTAGAGGGAACTC
NV3-TGA SDM	R	GAGTTCCCTCTAGGAGACTTATCCTTCAGTCAAGCACATGGAGGTAGGG
G 1415 real-time	F	TGGGAGGGGCCTTCCTTCTG
NV real-time	F	ACCCAAGCAACTACCTCAAC
NV real-time	R	GGTCCTTAGATCCTCTGAGACT

The bold letters indicate restriction enzyme sites.

## Data Availability

Not applicable.

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
