# Peer review of "Effects of Non-Virion Gene Expression Level and Viral Genome Length on the Replication and Pathogenicity of Viral Hemorrhagic Septicemia Virus"

_viruses, 2022, doi:10.3390/v14091886_

Round 1
Reviewer 1 Report
This manuscript expands the knowledge of previous work by the authors and others regarding the function of the NV gene in the novirhabdovirus VHSV. The methods are well done regarding the viral recombination while the results support the authors’ conclusions.
The paper is recommended for publication after the authors’ address a few minor suggestions:
1. Table 1 is missing (lines 108, 121, 127, 177). Or should Table 1 be reworded as Figure 1 which lists the different recombinants as a helpful reference for the reader.
2. Line 268. Explain why CPE progression might occur rarely or at all – there is some explanation here but it does not speculate why the CPE occurs sometimes rather than consistently not at all. Please add a sentence or two to discuss further.
3. Lines 343-345. A part of the sentence here appears to be missing. Please reword.
Reviewer 2 Report
VHS fish virus corrections July 31, 2022
I enjoyed reading this paper, and it is a nice study, which can be improved with some attention to details.
English is incorrect in several places. I have marked the manuscript extensively.
The correct plural of more than one species of fish is “fishes”; needs to be corrected throughout
The authors tend to cite just their own work, this is not ok at present.
The authors need to clearly differentiate this paper from their other very similar papers.
It is very important to more clearly related to others work in the field and to cite more recent papers.
Too many “we”s throughout, we, we, we……
Try to avoid “We” at the start of each sentence and each paragraph.
Eliminate 50-70% of them (I marked and changed many of these on the marked copy).
Word “harbor” is mis-used in many places, substitute “contain” or “have/having” or “possess”.
To many “usings” too.
Should have line numbers.
In Some cases, evolutionary principles are not used correctly, implies a “will” to the virus, which is not the case at all; this is all natural selection.
et al. needs a period throughout.
Graphs all need units, also a key to what asterisks * mean; as well as are the bars standard error, range, or standard deviation; also, how was this measured.
Results are mixed in with Discussion. They should be more clearly separated. Do not restate the results in the discussion, but interpret them instead in light of the literature.
Please temper the longer genome length interpretation, as this is merely a hypothesis at this point.
Change VHSV or VHSv to be consistent throughout the paper, including the figures, tables, and their captions.
Please clarify % mortality vs. mortality rate in the Discussion.
Do not use “didn’t” in professional scientific writing.
Literature is rather dated, there are some more recent papers, which can be cited.
e.g., Baillon et al., Journal of General Virology 2017;98:1181–1184 DOI 10.1099/jgv.0.000830A single amino acid change in the non-structural NV protein impacts the virulence phenotype of Viral hemorrhagic septicemia virus in trout
see https://www.sciencegate.app/keyword/676684
Niner, Megan D., Carol A. Stepien, Bartolomeo Gorgoglione, and Douglas W. Leaman. Genomic and Immunogenic Changes of Piscine Novirhabdovirus (Viral Hemorrhagic Septicemia Virus) over Its Evolutionary History in the Laurentian Great Lakes. Edited by Maria del Mar Ortega-Villaizan. PLOS ONE 16, no. 5 (May 28, 2021): e0232923. doi:10.1371/journal.pone.0232923.
